# Does Internet use alleviate household financial vulnerability? An empirical analysis based on panel data from China Family Panel Studies (CFPS)

Zeliang Yu[1,2], Heyu Li [ID][2]*, Pei Guo[1], Lin He[2]

**1** College of Economics and Management, China Agricultural University, Beijing, China, **2** College of Management, Zhongkai University of Agriculture and Engineering, Guangzhou, Guangdong, China

* heyu.li@foxmail.com

## Abstract

As digital technologies increasingly shape household financial behavior, understanding their role in buffering financial shocks has become an important policy and research concern. Drawing on panel data from the 2016, 2018, and 2020 waves of the China Family Panel Studies (CFPS), this paper investigates whether and how Internet use mitigates household financial vulnerability. The empirical results indicate that Internet use significantly lowers both the likelihood and severity of household financial vulnerability, a conclusion that holds after a series of robustness tests and corrections for endogeneity. Mechanism analysis reveals that this effect operates primarily through three channels: boosting income growth, promoting wealth accumulation, and enhancing risk management capabilities. Further heterogeneity analysis shows stronger mitigating effects in western regions, rural areas, and among households headed by individuals under the age of 60. These findings suggest several policy implications: deepen digital development by improving infrastructure and raising digital literacy to broadly reduce financial vulnerability; expand household income and wealth accumulation channels through innovative, technology-driven financial products that strengthen risk management; and prioritize equitable digital advancement, focusing on western regions, rural households, and older populations to narrow the digital divide and reduce systemic financial risk.

## 1. Introduction

As fundamental socio-economic units, households face financial risks that not only affect their welfare and sustainable development but may also spill over into banks, enterprises, government sectors, and other domains, potentially causing systemic financial risks detrimental to high-quality economic growth [1]. Since the global financial crisis of 2008, household financial risks have garnered extensive attention [2–3]. In this

**Data availability statement:** The data underlying the results of this study are available from the China Family Panel Studies (CFPS), administered by the Institute of Social Science Survey (ISSS) at Peking University. Researchers can apply for access at: https://cfpsdata.pku.edu.cn/#/home or contact isss.cfps@pku.edu.cn.

**Funding:** 1. National Key R&D Program "Intergovernmental International Science and Technology Innovation Cooperation" project "Natural Solutions-based Agricultural Nutrient Management and Sustainable Transformation between China and the European Union" (2023YFE0105000); 2. National Natural Science Foundation Youth Project "Research on Resilience Governance of Poverty-stricken Farmers in Western China under the Impact of Natural Disasters" (72403234); 3. Guangdong Provincial Federation of Social Sciences "Zhongkai Agricultural and Engineering University Guangdong-Hong Kong-Macao Greater Bay Area Rural Finance and Agricultural Investment Research Center" project (Guangdong Federation of Social Sciences Letter [2022] No. 5). The funders had no role in study design, data collection and analysis, decision to publish, or preparation of the manuscript.

**Competing interests:** The authors have declared that no competing interests exist.

context, understanding and mitigating household financial vulnerability becomes crucial for sustaining economic stability and promoting inclusive development.

The concept of financial vulnerability offers an innovative lens for examining household financial risk, focusing not only on exposure to shocks but also on the resilience capacity of households themselves. Typically, household financial vulnerability is jointly determined by financial factors such as household income, expenditures, assets, and liabilities [4–5]. Multiple pieces of evidence suggest the pressing nature of household financial vulnerability in China. Firstly, driven by housing financialization and credit shocks, total household loans in China have continuously increased, with rising household debt-to-income ratios, asset-liability ratios, and household leverage, thereby intensifying household financial risks [6–7]. Secondly, influenced by risk expectations and financial literacy, Chinese households primarily allocate assets into non-financial forms, reflecting limited participation in financial markets. Housing constitutes the dominant share of household assets, with severe under allocation of liquid assets, which constrains households' debt-servicing capacities and weakens their financial resilience [8]. Thirdly, factors such as high social expenditure, housing costs, healthcare, and education expenses collectively exacerbate financial vulnerabilities by expanding households' financial risk exposure [9].

Parallel to these financial trends, the digital revolution is profoundly reshaping household economic behaviors. A growing body of literature demonstrates that digital technologies, particularly Internet use, can increase household income, optimize consumption and asset allocation, and alleviate credit constraints [10]. For instance, studies show that Internet use enhances human capital and labor market competitiveness [11], facilitates entrepreneurship and business income [12], and improves access to financial information and products, thereby promoting wealth accumulation [13]. However, the net effect of Internet use on overall financial vulnerability remains ambiguous and inadequately explored. The multifaceted impact of digital technology introduces complexity, and the persistent digital divide in China, evolving from disparities in access to disparities in usage efficacy [14], which raises a critical question: can disadvantaged groups effectively leverage digital tools to mitigate their financial vulnerability, or will these technologies exacerbate existing inequalities?

A thorough review of the existing literature reveals several critical gaps that this study aims to address. Firstly, while prior studies have examined isolated channels such as income [15] or insurance participation [16], none have integrated the multiple pathways of income growth, wealth accumulation, and risk management into a coherent theoretical framework to explicate how Internet use impacts financial vulnerability. Secondly, methodological limitations persist. Many studies fail to adequately address endogeneity concerns stemming from reverse causality and omitted variables, casting doubt on the reliability of the estimated effects [17]. Furthermore, there is a notable scarcity of research exploring the heterogeneous effects of Internet use across diverse demographic and regional subgroups. Lastly, the measurement of financial vulnerability often relies on oversimplified binary or categorical indicators [8,18], which fails to capture the continuous and gradational nature of the phenomenon, potentially obscuring nuanced relationships.

To bridge these gaps, this study draws on panel data from China Family Panel Studies (CFPS) for 2016, 2018, and 2020. We construct a continuous and precise measure of household financial vulnerability grounded in the concepts of financial margin and solvency [19–20]. To address endogeneity and establish causal inference, we employ a rigorous empirical framework that integrates panel Logit and Tobit models with instrumental variable (IV) techniques, enabling us to assess the effect of Internet use on both the likelihood and the extent of financial vulnerability. Beyond a single explanatory channel, we theoretically develop and empirically test a mediating framework encompassing income growth, wealth accumulation, and risk management optimization. Finally, we conduct heterogeneity analyses to examine whether the effects vary across regions, urban-rural settings, and household-head age groups.

The marginal contributions of this study are threefold. Methodologically, we advance the measurement of financial vulnerability and employ robust causal inference strategies to mitigate endogeneity, enhancing the credibility of our findings. Theoretically, we develop and validate a multi-channel framework that provides a holistic understanding of the mechanisms linking digital technology to financial resilience. From a policy perspective, our heterogeneity analysis offers nuanced evidence to support the design of targeted, equitable digital inclusion policies aimed at reducing financial vulnerability, particularly among western regions, rural households, and younger populations.

Our results confirm that Internet use significantly reduces both the likelihood and the severity of household financial vulnerability. This effect is channeled through increased household income, enhanced wealth accumulation, and improved risk management capabilities. The mitigating effects are found to be more pronounced in western regions, rural areas, and among households with heads under the age of 60.

## 2. Theoretical analysis and research hypotheses

This study is grounded in an integrated theoretical framework that combines perspectives from the digital divide theory and household finance theory. The digital divide theory posits that inequality in access to, use of, and outcomes from information and communication technologies can exacerbate existing socioeconomic disparities [21]. Conversely, bridging this divide can foster inclusion and empowerment. This theory guides our analysis of the heterogeneous effects of Internet use across different groups.

Concurrently, our research is informed by the household finance theory [22], which examines how households use financial instruments to achieve their life goals, emphasizing the roles of information asymmetry, transaction costs, and risk management. The convergence of these two theoretical streams provides a powerful lens for analyzing how digital technology, as a disruptive force, influences household-level financial outcomes by altering information sets, reducing market frictions, and reshaping risk management capabilities. Fig. 1 presents a flowchart of the mechanism analysis, highlighting the pathways through which Internet use alleviates household financial vulnerability.

### 2.1. Internet use and household financial vulnerability

With the progressive improvement of information infrastructure and rapid advancement of digital technology, widespread Internet use has significantly transformed the macro-financial environment and enhanced household information acquisition efficiency [23]. Nowadays, Internet use by households for learning, work, business activities, social interaction, and entertainment has become increasingly prevalent, thereby reducing household financial vulnerability. As delineated by our integrated framework, this pervasive use is theorized to reduce household financial vulnerability through several distinct channels.

First, Internet-based learning promotes effective dissemination of knowledge, enhancing household financial literacy [24], thus enabling better financial decision-making and reducing financial risk vulnerability. Second, Internet use enhances human capital through online learning and skill acquisition, thereby improving labor market competitiveness and household income [25], thereby reducing susceptibility to financial shocks due to income inadequacy. Third, acting as a powerful tool to mitigate information asymmetry and lower transaction costs, Internet use facilitates entrepreneurial

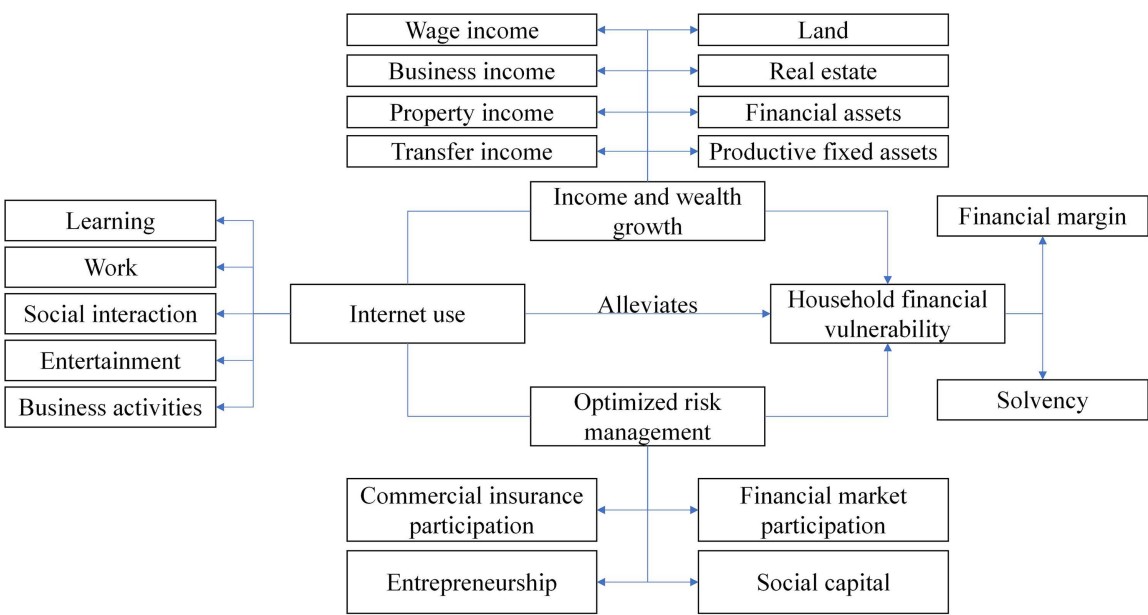

**Fig 1. Flowchart of the mechanism analysis.**

activities and market efficiency [14]. Moreover, Internet use facilitates the accumulation of social capital, which provides access to economic opportunities, resources, and support networks, thereby enhancing financial resilience [26]. Lastly, Internet-facilitated entertainment improves physical and mental health by enhancing social adaptation, life diversity, and healthy living habits [27], thereby reducing medical expenditure uncertainties and mitigating health-related financial risks.

Synthesizing these arguments derived from household finance and digital divide theories, we propose that Internet use serves as a multifaceted tool that empowers households to mitigate financial risks. Thus, we postulate the following hypothesis.

**Hypothesis 1:** Internet use reduces household financial vulnerability.

## 2.2. Mechanisms of Internet use affecting household financial vulnerability

Fundamentally, reducing household financial vulnerability necessitates enhancing risk management capabilities via increased household income and wealth accumulation. Concerning income growth, Internet use generally elevates household income [28], particularly benefiting rural households after significant narrowing of the urban-rural digital divide [29–30].

Specifically, Internet use increases wage, business, asset, and transfer income by improving information access, enhancing human capital, expanding social networks, increasing financial service accessibility, entrepreneurial support, and transaction cost reduction [26,31,32]. Regarding wealth accumulation, Internet use also significantly contributes to household wealth growth, particularly in the expansion of risky financial assets. This mechanism operates in two primary ways. Firstly, Internet use reduces digital information search costs, enabling households to access financial market information more efficiently [33–34]. Secondly, the educational and information dissemination functions of the Internet enhance household financial literacy, leading to more sophisticated and rational wealth management and investment decisions [34]. Through these pathways, Internet use not only boosts household wealth accumulation, particularly in financial asset allocation, but also strengthens financial resilience against potential risks, thereby mitigating financial vulnerability.

The foregoing discussion outlines the theoretical pathways through which Internet use augments income and wealth. Based on this, we hypothesize that:

**Hypothesis 2:** Internet use mitigates household financial vulnerability through increased household income and wealth accumulation.

A lack of effective risk management strategies significantly contributes to household financial vulnerability [5]. Effective risk management assists households in managing unexpected events and financial crises, thus broadly reducing the likelihood of financial distress. Participation in insurance markets, risky financial markets, entrepreneurial activities, and social capital accumulation represent critical pathways for mitigating household financial vulnerability [9,35–37]. Internet use significantly enhances these pathways.

Firstly, Internet use considerably increases households' probability of participating in insurance markets by improving financial literacy, reducing transaction costs, enhancing accessibility to insurance products, and promoting social interaction [16]. Families can conveniently access comprehensive information regarding insurance products, understand their scope and associated risks to make informed decisions, thereby reducing exposure to uncontrollable risks and enhancing resilience.

Secondly, Internet use rectifies cognitive biases regarding financial risks, improves investment convenience, and provides richer financial information, thereby enhancing financial decision-making and increasing households' engagement with risky financial markets [38–39]. For example, the Internet facilitates accurate assessments of returns and risks associated with financial assets like stocks and mutual funds, leading to more diversified asset allocation and reducing the phenomenon of limited market participation. Through participation in risky financial markets via the Internet, households can achieve higher investment returns, enhance wealth accumulation, and improve resilience against financial shocks [40].

Thirdly, Internet use significantly supports entrepreneurship by increasing information efficiency, broadening access to financial information, and strengthening entrepreneurial skills [31]. Internet access provides potential entrepreneurs with greater commercial opportunities, market insights, and financing channels, promoting entrepreneurial activities. Through entrepreneurship, households diversify income streams, accumulate wealth, and consequently improve their ability to manage financial shocks.

Finally, widespread Internet use enhances households' social capital [41]. Technological advancements improve connectivity and facilitate social interactions, enabling households to accumulate valuable social resources [42]. Such resources include economic support, emotional assistance, and information sharing, thereby ensuring better external support and reducing isolation during financial crises.

The discussed mechanisms illustrate how Internet use optimizes household risk management practices across multiple dimensions. Therefore, we derive the following hypothesis.

**Hypothesis 3:** Internet use mitigates household financial vulnerability by enhancing risk management practices.

## 3. Research design

### 3.1. Data sources

The data utilized in this study are primarily derived from the China Family Panel Studies (CFPS) conducted by the Institute of Social Science Survey at Peking University in 2016, 2018, and 2020, covering samples from 25 provinces across China. This dataset primarily describes the dependent variable (household financial vulnerability), the key independent variable (Internet use), and control variables at household and community levels. Regional control variables are mainly sourced from the China Statistical Yearbook. After excluding observations with missing key variables and households whose heads were under the age of 16, the final samples for 2016, 2018, and 2020 include 13,214, 11,602, and 6,776 households, respectively. These observations form an unbalanced panel dataset to preserve data integrity.

## 3.2. Variable selection

### 3.2.1. Dependent variable: household financial vulnerability.
Since its introduction in the early 20th century [43], the concept of vulnerability has been widely applied in fields such as ecology, economics, and livelihoods and was introduced into micro-level household studies in the late 20th century [44]. Current measures of household financial vulnerability primarily involve two dimensions: objective and subjective [45]. Regarding objective measures, although no unified consensus has been established, the development of measurement approaches has generally undergone three stages.

Initially, household financial vulnerability was treated equivalently to poverty vulnerability. The second stage measured household financial vulnerability through debt indicators with set thresholds, such as debt-to-income ratio, debt-to-asset ratio, outstanding debt-to-income ratio, and interest expenses-to-disposable income ratio [19,46,47], although these thresholds should dynamically adjust to socioeconomic changes. The third stage emphasizes the household's capacity to manage risks, such as asset liquidity [19], introducing concepts like financial margins and solvency [20]. This study constructs a household financial vulnerability index based on these concepts.

Firstly, financial margin is introduced to assess whether a household experiences financial vulnerability: Financial margin = total income + current assets – regular expenditures – unexpected expenditures. A financial margin below zero indicates financial vulnerability (coded as 0); otherwise, it is coded as 1.

Secondly, when financial vulnerability is coded as 0, solvency is used to measure the degree of vulnerability: Solvency = (Total income + easily liquefiable assets)/ (Unexpected expenditures + regular expenditures). The degree of household financial vulnerability is the inverse of solvency, with values greater than 1. If financial vulnerability is coded as 1, the degree of vulnerability is set to 1 (boundary value). Higher values indicate greater financial vulnerability. The vulnerability degree is logarithmically transformed for analysis.

This financial margin and solvency approach was selected over simpler debt-ratio measures because it provides a more holistic view of a household's ability to withstand financial shocks, capturing both liquidity through current assets and income-based resilience. This aligns with the core principles of household finance theory, which emphasizes assessing risk management capabilities and buffer stocks against adverse events [5,19].

### 3.2.2. Independent variable: internet use.
While digital technologies have multiple applications, Internet use remains the most common among households. This study measures Internet use primarily through frequency, as the intensity of use is theorized to be a key determinant of its impact on human capital accumulation, information acquisition efficiency, and social capital formation [15,42]. CFPS data collect information on Internet use frequency for learning, work, social interaction, entertainment, and business activities among household members. Responses include nearly every day, 3–4 times a week, 1–2 times a week, 2–3 times a month, once a month, once every few months, and never. Frequencies are scored from highest to lowest (6–0), and the sum across these five dimensions provides the overall Internet use level.

A frequency-based composite index captures the breadth and depth of household engagement with the digital economy more effectively than a simple binary access measure. This operationalization allows us to test for a potential dose-response relationship between digital immersion and reduced financial vulnerability, a central premise of digital divide theory which emphasizes that usage intensity, not merely access, determines socioeconomic outcomes [48].

### 3.2.3. Control variables.
The selection of control variables is guided by established economic theory and regional economic theory and prior literature on household financial vulnerability. We include a comprehensive set of covariates that are theoretically pertinent and have been empirically shown to correlate with a household's financial health, ensuring that we isolate the net effect of Internet use by accounting for potential confounding factors.

The first group includes household-head characteristics: gender (male = 1, female = 0), age, education (illiterate = 0, primary = 1, junior high = 2, senior high = 3, college and above=4), health status (excellent = 1, very good = 2, good = 3, fair = 4, poor = 5), marital status (married living with spouse = 1, others = 0), and household registration type (rural = 0, urban = 1). Gender, age, education, health status: These variables are core components of human capital theory [49]. They

fundamentally influence earning potential, health expenditure risks, and financial decision-making capability, all of which are direct determinants of financial vulnerability [9,20]. For instance, lower education and poor health are consistently associated with higher financial fragility. Marital status: Marital status is a key indicator of household economic structure and intra-household risk-sharing capacity, affecting income pooling, consumption economies of scale, and expenditure needs [50]. Household Registration type: The Hukou system in China creates a fundamental institutional divide, determining access to public services, formal credit markets, and urban labor markets, thereby directly impacting financial stability and vulnerability [51].

The second group encompasses household characteristics: household size, number of minors (individuals under 16), and mortgage status (no mortgage = 0, mortgage = 1). Household size, number of minors: These factors capture demographic dependencies and lifecycle-related expenditure pressures such as education, childcare costs that strain household budgets and increase financial fragility, as outlined in household production theory and models of household consumption [52]. Mortgage status: Mortgage debt represents a major fixed financial obligation and is a primary source of financial leverage and risk for households, significantly influencing their susceptibility to income shocks [2,7].

The third group consists of regional characteristics: region (eastern = 1, central = 2, western = 3), provincial economic development (logarithm of per capita GDP), and social environment. Region (East/Central/West): China's regions exhibit significant disparities in economic development, financial market depth, and social safety nets, creating geographically unequal exposures to macroeconomic risks and opportunities [26]. Provincial economic development: Macroeconomic conditions at the provincial level influence employment opportunities, wage levels, and asset prices, all of which are fundamental contextual factors affecting household finances [1]. Social environment: Perceptions of the social environment—such as social trust, income inequality, and government integrity—shape households' risk perceptions, access to informal insurance mechanisms, and overall sense of economic security, thereby influencing their financial vulnerability [53].

Some variables were logarithmically transformed to minimize discrepancies in magnitude and ensure comparability. Detailed descriptive statistics for each variable are presented in Table 1 below.

### 3.3. Methods

**3.3.1. Benchmark regression.** To examine the impact of Internet use on household financial vulnerability, we select two econometric models tailored to the distinct nature of our dependent variables. Our model selection is driven by the necessity to obtain consistent and unbiased estimates while addressing the specific statistical challenges posed by each measure of financial vulnerability.

Firstly, the variable indicating whether a household is financially vulnerable is binary. Employing a classic linear probability model (LPM) for such a dichotomous outcome would be inappropriate, as it can predict probabilities outside the [0,1] logical range and produce heteroscedastic errors, leading to inefficient and biased inferences. To overcome these limitations and, crucially, to control for unobserved time-invariant household heterogeneity, we adopt a fixed-effects panel Logit model. This model is specified as Equation (1).

To examine the impact of Internet use on household financial vulnerability, two econometric models are selected.

$$\text{Prob}(FV_{it} = 1) = \alpha_0 + \alpha_1 \text{Digital}_{it} + \alpha_2 Z_{it} + \text{year}_t + c_i + \varepsilon_{it} \tag{1}$$

Secondly, our measure for the degree of financial vulnerability is a continuous variable that is left-censored at the value of 1 (households not vulnerable are assigned this minimum value). Applying ordinary least squares (OLS) regression to such a censored dependent variable would result in inconsistent and biased estimates, as OLS would underestimate the effect of covariates for observations at the censorship point [54]. Therefore, we employ a panel Tobit model, which is explicitly designed to handle censored data and provide consistent parameter estimates. The model is specified as Equation (2).

**Table 1. Variable descriptions and descriptive statistics.**

| Variable Type | Variable Name | Observations | Min | Max | Mean | Std. Dev. |
|---|---|---|---|---|---|---|
| Dependent Variable | Household Financial Vulnerability (Binary) | 31592 | 0 | 1 | 0.643 | 0.479 |
| | Household Financial Vulnerability Degree | 31592 | 0 | 9.736 | 0.311 | 0.691 |
| Independent Variable | Internet Use | 31592 | 0 | 30 | 12.103 | 9.914 |
| Control Variables | Gender of Household-head | 31579 | 0 | 1 | 0.530 | 0.499 |
| | Age of Household-head | 31592 | 16 | 95 | 50.731 | 14.666 |
| | Education of Household-head | 31564 | 0 | 4 | 1.653 | 1.260 |
| | Health Status of Household-head | 31514 | 1 | 5 | 3.142 | 1.212 |
| | Marital Status of Household-head | 31523 | 0 | 1 | 0.841 | 0.365 |
| | Hukou (Household Registration) Type | 31362 | 0 | 1 | 0.741 | 0.438 |
| | Household Size | 31592 | 1 | 21 | 3.765 | 1.925 |
| | Number of Minors | 31592 | 0 | 7 | 0.396 | 0.715 |
| | Mortgage Status | 31592 | 0 | 1 | 0.105 | 0.307 |
| | Region | 31592 | 1 | 3 | 1.960 | 0.834 |
| | Economic Development Level | 31592 | 10.219 | 12.009 | 10.878 | 0.402 |
| | Social Environment | 29804 | 0 | 80 | 50.267 | 15.459 |

$$Degree(FV_{it}) = \beta_0 + \beta_1 Digital_{it} + \beta_2 Z_{it} + year_t + c_i + \varepsilon_{it} \tag{2}$$

In the equations above, $FV_{it}$ denotes the dependent variable, representing either whether household i in year t experiences financial vulnerability or the degree of vulnerability; $Digital_{it}$ represents Internet use of household i in year t; $Z_{it}$ is a set of control variables including household-head, household, and regional-level characteristics; $c_i$ denotes provincial fixed effects to eliminate unobserved regional heterogeneity; and $\varepsilon_{it}$ are random error terms.

Although the coefficient estimates from the Tobit model are consistent, their standard errors may be biased if the error term is heteroscedastic. To test the robustness of our inference on the degree of financial vulnerability, we employed boot-strapped standard errors with 500 replications to assess the potential impact of heteroscedasticity. The results confirmed that the significance and direction of all key coefficients remained unchanged, and the magnitude of the bootstrapped standard errors was very similar to our original estimates. This indicates that heteroscedasticity does not pose a substantial threat to the validity of our baseline conclusions. Therefore, for the sake of consistency and presentation clarity, we maintain the original standard errors in our reported results.

Prior to conducting the multivariate regression analysis, we assessed the potential issue of multicollinearity among all independent variables by calculating the Variance Inflation Factor (VIF). The results indicate that the mean VIF value is 1.44, and all individual VIF values are significantly below the common threshold of 10, with the highest being 2.13 for the variable Region. This confirms that severe multicollinearity is not a concern in our model and would not bias the estimation results.

**3.3.2. Endogeneity treatment.** The benchmark regression models may suffer from endogeneity problems such as omitted variable bias and reverse causality, leading to biased estimates of the coefficient on Internet use. First, reverse causality may exist, as financially vulnerable households or households with higher vulnerability degrees might reduce expenditures related to the Internet, leading to lower frequency and degree of digital technology adoption. Second, omitted variable bias may arise; although the models include extensive controls from the household-head, household, and regional levels, some unobserved and unidentifiable factors could still affect both Internet use and financial vulnerability simultaneously.

To alleviate these potential endogeneity issues, this study employs instrumental variable approaches including IV-Probit and IV-Tobit models. Formally, these IV model specifications are as follows.

IV-Probit model:

$$\text{Digital}_{it} = \alpha_0 + \alpha_1 \text{Instrument}_{it} + \alpha_2 \, Z_{it} + \varepsilon_{it} \tag{3}$$

$$\text{Prob}(FV_{it} = 1) = \alpha_0 + \alpha_1 \text{Instrument}_{it} + \alpha_2 \, \text{Digital}_{it} + \alpha_3 \, Z_{it} + \varepsilon_{it} \tag{4}$$

IV-Tobit model:

$$\text{Digital}_{it} = \beta_0 + \beta_1 \text{Instrument}_{it} + \beta_2 \, Z_{it} + \varepsilon_{it} \tag{5}$$

$$\text{Degree}(FV_{it}) = \beta_0 + \beta_1 \text{Instrument}_{it} + \beta_2 \, \text{Digital}_{it} + \beta_3 \, Z_{it} + \varepsilon_{it} \tag{6}$$

**3.3.3. Mechanism analysis.** To test the mediating roles of income growth, wealth accumulation, and risk management in the relationship between Internet use and household financial vulnerability, the following econometric model is established [55].

$$M_{it} = \gamma_0 + \gamma_1 \text{Digital}_{it} + \gamma_2 \, Z_{it} + \varepsilon_{it} \tag{7}$$

In Equation (7), $M_{it}$ represents the mechanism variable (mediating variable). If the coefficient $\gamma_1$ is statistically significant and its sign aligns with theoretical expectations, combined with evidence from existing literature demonstrating the impact of $M_{it}$ on household financial vulnerability, it indicates that Internet use indeed influences household financial vulnerability through the $M_{it}$.

**3.3.4. Robustness checks.** To ensure the reliability and credibility of our main findings, we will employ a series of robustness checks in the subsequent analysis section. Specifically, we will adopt the following three strategies: replacing the measurement of the dependent variable (household financial vulnerability) with alternative proxy measures; replacing the core independent variable (Internet use) with alternative measuring approaches; and employing the Heckman two-stage model to correct for potential sample selection bias. The results of these robustness checks are presented in the following section.

## 4. Results

### 4.1. Benchmark regression analysis

We rigorously evaluated the validity of our control variable selection and the robustness of our empirical specification. Spearman correlation analysis confirmed that all control variables are significantly correlated with both Internet use and financial vulnerability in theoretically expected directions. Furthermore, coefficient stability tests demonstrated that the estimated effect of Internet use remains positive, statistically significant at the 1% level, and remarkably stable in magnitude across multiple model specifications that incrementally incorporate different sets of control variables. Detailed results of these analyses are available upon request.

The benchmark regression results presented in Table 2 indicate that Internet use significantly alleviates household financial vulnerability. Specifically, Internet use reduces both the likelihood that households fall into financial vulnerability and the degree of such vulnerability. The relationships between other control variables and household financial vulnerability are summarized as follows:

Regarding household-head characteristics, households headed by males exhibit lower financial vulnerability compared to those headed by females. Furthermore, an increase in the household-head's age, educational level, and health

**Table 2. Benchmark regression results.**

| Variables | Household Financial Vulnerability (Binary) | Household Financial Vulnerability Degree |
|---|---|---|
| Internet Use | 0.013*** | −0.022*** |
| | (0.003) | (0.001) |
| Gender of Household-head | 0.111** | −0.097*** |
| | (0.055) | (0.021) |
| Age of Household-head | 0.007** | −0.007*** |
| | (0.003) | (0.001) |
| Education of Household-head | 0.060 | −0.136*** |
| | (0.037) | (0.011) |
| Health Status of Household-head | −0.050** | 0.072*** |
| | (0.020) | (0.009) |
| Marital Status of Household-head | −0.234** | −0.078** |
| | (0.093) | (0.030) |
| Hukou (Household Registration) Type | 0.148 | 0.307*** |
| | (0.118) | (0.028) |
| Household Size | 0. 117*** | −0.032*** |
| | (0.021) | (0.007) |
| Number of Minors | −0.032 | 0.047** |
| | (0.034) | (0.016) |
| Mortgage Status | −0.663*** | 0.449*** |
| | (0.080) | (0.032) |
| Region | 0.390** | 0.038** |
| | (0.196) | (0.019) |
| Economic Development Level | 1.616*** | −0.497*** |
| | (0.142) | (0.040) |
| Social Environment | −0.001 | 0.003*** |
| | (0.001) | (0.001) |
| Provincial Fixed Effects | Controlled | Controlled |
| Year Fixed Effects | Controlled | Controlled |
| N | 11913 | 29551 |
| Prob > chi2 | 0.0000 | 0.0000 |

*, **, and *** indicate statistical significance at the 10%, 5%, and 1% levels, respectively. Standard errors are reported in parentheses below the coefficients.

status is beneficial for reducing household financial vulnerability. Households headed by individuals without spouses are more likely to experience financial vulnerability; however, their degree of financial vulnerability is comparatively lower. Compared to households with rural Hukou, those with urban Hukou tend to experience a higher degree of financial vulnerability.

In terms of household characteristics, larger household sizes contribute to alleviating financial vulnerability, whereas a greater number of minors within a household tends to increase the degree of vulnerability. Additionally, households with mortgage debt show increased financial vulnerability. For regional characteristics, households located in eastern and central regions, compared to those in western regions, have a higher probability of experiencing financial vulnerability, although their degree of vulnerability is relatively lower. A higher regional economic development level helps reduce household financial vulnerability, while a poorer social environment contributes to increasing vulnerability.

To account for potential cross-sectional dependence, we employ cluster-robust standard errors. Since a small number of households changed provinces across survey waves, we cluster standard errors at the province of residence in the first observed period for each household to ensure that the clustering structure is nested within panels [56]. Reassuringly, Internet use remains highly statistically significant (p < 0.01) in both alternative specifications. The signs of the coefficients align with our baseline nonlinear models: Internet use is associated with a higher probability of falling into financial vulnerability yet mitigates the severity of vulnerability for those already exposed. This pattern of results, which holds under a more conservative estimation strategy designed to account for cross-sectional dependence, strengthens our confidence in the robustness of the core conclusions.

## 4.2. Robustness test

### 4.2.1. Replacing dependent variables.
As previously mentioned, existing studies have employed multiple methods to measure household financial vulnerability, such as utilizing "insolvency" and "income shortfall" indicators to reflect current and potential household financial vulnerabilities respectively [36]. Specifically, "insolvency" refers to a situation where total household assets are less than total liabilities, while "income shortfall" indicates that household income is insufficient to cover expenses. Furthermore, household financial vulnerability levels are categorized into three groups: 1 denotes no vulnerability, 2 indicates a lower degree of vulnerability, and 3 signifies a higher degree of vulnerability. A fixed-effects panel linear regression model is applied to examine the impact of Internet use on these financial vulnerability classifications.

The results, presented in Table 3, reveal that Internet use significantly reduces the likelihood of households experiencing "insolvency" and "income shortfall" scenarios, as well as decreasing the degree of household financial vulnerability. Thus, the benchmark regression findings are robust.

### 4.2.2. Replacing independent variables.
The digital divide manifests in two dimensions: the "access divide" and the "usage divide". Some researchers argue that the "access divide" persists [29]. Accordingly, this study examines whether households experience an "access divide" using two indicators: "mobile Internet access" and "computer-based Internet access". These alternative independent variables are tested to determine their impact on the occurrence and degree of household financial vulnerability.

The results in Table 4 demonstrate that, after replacing the independent variable, Internet use significantly reduces the probability of households falling into financial vulnerability and decreases the degree of such vulnerability. These results further substantiate that digital technology applications indeed mitigate household financial vulnerability.

**Table 3. Robustness test: alternative dependent variables.**

| Variables | Household Financial Vulnerability (Binary) | | Household Financial Vulnerability Degree (Three Categories) |
| --- | --- | --- | --- |
| | Income Shortfall | Income Shortfall | |
| Internet Use | 0.011*** | 0.014** | −0.004*** |
| | (0.003) | (0.006) | (0.001) |
| Control Variables | Controlled | | |
| Provincial Fixed Effects | Controlled | | |
| Year Fixed Effects | Controlled | | |
| N | 14045 | 2408 | 29551 |
| Prob > chi2 | 0.0000 | 0.0000 | 0.0000 |

*, **, and *** indicate statistical significance at the 10%, 5%, and 1% levels, respectively. Standard errors are reported in parentheses below the coefficients.

**4.2.3. Replacing model framework.** Given that households as economic actors have inherent self-selection behaviors, there may be potential endogeneity issues arising from sample self-selection bias. To address this, the Heckman two-stage model is adopted to control for endogeneity due to non-random household selection.

The regression results presented in Table 5 consistently show that Internet use significantly alleviates household financial vulnerability, confirming the robustness of the main empirical findings.

## 4.3. Endogeneity discussion: instrumental variable approach

As discussed, potential endogeneity may compromise the identification of a causal effect. To address this concern, we employ an instrumental variable (IV) approach. A valid instrument must satisfy two conditions: relevance (correlated with the endogenous regressor, Internet use) and exclusion (uncorrelated with the error term in the main equation, thereby influencing financial vulnerability only through Internet use).

We select the straight-line geographic distance from each household's provincial capital to Hangzhou as the instrument. We justify this choice on two grounds. First, regarding relevance, Hangzhou is a flagship city in China's digital economy, housing tech giants like Alibaba and serving as a hub for digital technology innovation and policy. Consequently, provinces farther from Hangzhou may experience delayed diffusion of digital infrastructure and expertise, leading to lower average levels of household Internet adoption. This negative correlation is confirmed in our first-stage results. Second, concerning the exclusion restriction, it is highly plausible that this geographic distance is exogenous to individual households' financial vulnerability. While provincial-level economic development might confound this relationship, we explicitly

**Table 4. Robustness test: alternative independent variables.**

| Variables | Household Financial Vulnerability (Binary) | Household Financial Vulnerability Degree (Three Categories) |
|---|---|---|
| Internet Use (Binary) | 0.174 *** | −0.253*** |
| | (0.049) | (0.023) |
| Control Variables | Controlled | |
| Provincial Fixed Effects | Controlled | |
| Year Fixed Effects | Controlled | |
| N | 29734 | 10393 |
| Prob > chi2 | 0.0000 | 0.0000 |

*\*, \*\*, and \*\*\* indicate statistical significance at the 10%, 5%, and 1% levels, respectively. Standard errors are reported in parentheses below the coefficients.*

**Table 5. Robustness test: alternative model framework.**

| Variables | First Stage | Second Stage |
|---|---|---|
| Internet Use | −0.016*** | −0.012*** |
| | (0.001) | (0.001) |
| Control Variables | Controlled | |
| N | 11913 | 29551 |
| Prob > chi2 | 0.0000 | 0.0000 |

*\*, \*\*, and \*\*\* indicate statistical significance at the 10%, 5%, and 1% levels, respectively. Standard errors are reported in parentheses below the coefficients.*

control for provincial GDP per capita, regional dummies and social environment. After controlling for these observed regional characteristics, the distance instrument is unlikely to affect household financial vulnerability through any other channel than its impact on Internet use accessibility and quality. The direct influence of historical distance on modern household financial decisions, after accounting for contemporary economic conditions, is arguably negligible.

The IV estimation results are presented in Table 6. The first-stage regressions show a statistically significant negative effect of distance on Internet use ($p < 0.05$), confirming the relevance of our instrument. Crucially, the first-stage F-statistics are 1135.08 and 1126.77 for the two models, vastly exceeding the Staiger-Stock rough threshold of 10 and the more stringent Stock-Yogo critical values. This provides strong evidence against a weak instrument problem, ensuring the reliability of our IV estimates.

In the second stage, the coefficients on Internet use remain statistically significant and maintain their expected signs. Specifically, Internet use significantly reduces the likelihood and the degree of household financial vulnerability at the 5% significance level. The direction and significance of these effects are consistent with our benchmark findings, but the IV estimates are larger in magnitude. This pattern is common when IV methods correct for the attenuation bias caused by measurement error or the downward bias from reverse causality in the OLS estimates. Furthermore, the Wald tests of exogeneity overwhelmingly reject the null hypothesis that Internet use is exogenous (p-values = 0.0000 and 0.0001), formally justifying the necessity of the IV approach.

In conclusion, the IV analysis corroborates our baseline findings, lending strong support to a causal interpretation of the mitigating effect of Internet use on household financial vulnerability. Hypothesis 1 is confirmed.

## 4.4. Mechanism analysis

Following the mechanism analysis approach [55], the previous section provided a theoretical framework analyzing how mechanisms such as income and wealth growth and optimized risk management influence household financial vulnerability. This section empirically tests only the relationships between the explanatory variable (Internet use) and these mediating mechanism variables.

**4.4.1. Income and wealth growth mechanism.** This study assesses whether Internet use reduces household financial vulnerability through the channels of increased household income or wealth accumulation, as shown in Table 7. Regarding the income growth mechanism, Internet use significantly and positively affects household wage income, demonstrating its role in increasing household income. Regarding the wealth growth mechanism, Internet use shows significantly positive impacts on household land holdings, real estate, financial assets, and productive

**Table 6. Instrumental variable estimation results.**

| Variables | Household Financial Vulnerability (Binary) | | Household Financial Vulnerability Degree | |
|---|---|---|---|---|
| | First Stage | Second Stage | First Stage | Second Stage |
| Internet Use | | 0.294** | | −0.359** |
| | | (0.136) | | (0.162) |
| Distance from Provincial Capital to Hangzhou | −0.000** | | −0.001** | |
| | (0.000) | | (0.000) | |
| Control Variables | Controlled | | | |
| N | 29734 | | 29551 | |
| First-stage F Statistic | 1135.08 | | 1126.77 | |
| Wald Exogeneity Test | χ2(1) = 16.64 [p = 0.0000] | | χ2(1) = 15.49 [p = 0.0001] | |

*\*, \*\*, and \*\*\* indicate statistical significance at the 10%, 5%, and 1% levels, respectively. Standard errors are reported in parentheses below the coefficients.*

**Table 7. Mechanism analysis: income and wealth growth.**

| Variables | Income Growth | | | | Wealth Growth | | | |
|---|---|---|---|---|---|---|---|---|
| | Wage Income | Business Income | Property Income | Transfer Income | Land | Real Estate | Financial Assets | Productive Fixed Assets |
| Internet Use | 0.062*** | 0.007 | −0.003 | 0.006 | 0.012** | 0.012** | 0.036*** | 0.009** |
| | (0.004) | (0.004) | (0.003) | (0.004) | (0.004) | (0.004) | (0.005) | (0.004) |
| Control Variables | Controlled | | | | | | | |
| Provincial Fixed Effects | Controlled | | | | | | | |
| Year Fixed Effects | Controlled | | | | | | | |
| N | 29734 | 29734 | 29734 | 29734 | 29734 | 29734 | 29734 | 29734 |

*, **, and *** indicate statistical significance at the 10%, 5%, and 1% levels, respectively. Standard errors are reported in parentheses below the coefficients.

fixed assets, suggesting that Internet use facilitates wealth accumulation. These findings support a clear positive transmission pathway: "Internet use→Income/Wealth Growth→Reduction of Household Financial Vulnerability". Therefore, Hypothesis 2 is confirmed.

**4.4.2. Optimized risk management mechanism.** This study further evaluates whether Internet use reduces household financial vulnerability through optimized risk management, considering dimensions such as commercial insurance participation, financial market participation, entrepreneurship, and social capital. Regression results presented in Table 8 indicate that Internet use significantly increases household participation in commercial insurance, financial markets, entrepreneurial activities, and social capital accumulation. These outcomes validate a clear positive transmission pathway: "Internet Use→Optimized Risk Management→Reduction of Household Financial Vulnerability". Thus, Hypothesis 3 is confirmed.

## 4.5. Heterogeneity analysis

Overall, Internet use effectively mitigates household financial vulnerability. In addition to the general benefits derived from digital technology, an essential consideration is whether there exist differentiated impacts regarding household financial vulnerability. This study investigates potential heterogeneous effects across three dimensions: region, urban-rural status, and household-head age, using subsample regressions and tests. The estimation results are presented in Table 9.

In terms of regional heterogeneity, for the binary measure of household financial vulnerability, the results for the central and western regions are consistent with the baseline regression, whereas the eastern region shows an insignificant impact. Moreover, households in more inland regions have a lower probability of financial vulnerability. For the continuous

**Table 8. Mechanism analysis: optimized risk management.**

| Variables | Commercial Insurance Participation | Financial Market Participation | Entrepreneurship | Social Capital |
|---|---|---|---|---|
| Internet Use | 0.022*** | 0.028*** | 0.018** | 0.014*** |
| | (0.004) | (0.002) | (0.006) | (0.003) |
| Control Variables | Controlled | | | |
| Provincial Fixed Effects | Controlled | | | |
| Year Fixed Effects | Controlled | | | |
| N | 8664 | 29702 | 3011 | 29734 |

*, **, and *** indicate statistical significance at the 10%, 5%, and 1% levels, respectively. Standard errors are reported in parentheses below the coefficients.

**Table 9. Heterogeneity analysis results.**

| Dependent Variable | | Regional Heterogeneity | | | Urban-Rural Heterogeneity | | Household-Head Age Heterogeneity | |
|---|---|---|---|---|---|---|---|---|
| | | Eastern | Central | Western | Rural | Urban | <60 | ≥60 |
| Household Financial Vulnerability (Binary) | Internet Use | 0.008 | 0.013** | 0.014** | 0.017*** | 0.006 | 0.014*** | 0.009 |
| | | (0.005) | (0.005) | (0.005) | (0.004) | (0.005) | (0.004) | (0.007) |
| | Prob > chi2 | 0.0000 | 0.0000 | 0.0000 | 0.0000 | 0.0000 | 0.0000 | 0.0000 |
| | N | 3599 | 3581 | 4610 | 6319 | 4964 | 8221 | 2453 |
| Household Financial Vulnerability Degree | Internet Use | −0.021*** | −0.018*** | −0.025*** | −0.025*** | −0.018*** | −0.021*** | −0.019*** |
| | | (0.002) | (0.002) | (0.002) | (0.002) | (0.002) | (0.001) | (0.003) |
| | Prob > chi2 | 0.0000 | 0.0000 | 0.0000 | 0.0000 | 0.0000 | 0.0000 | 0.0000 |
| | N | 10784 | 9011 | 9756 | 14848 | 14703 | 21253 | 8298 |
| Control Variables | | Controlled | | | | | | |
| Provincial Fixed Effects | | Controlled | | | | | | |
| Year Fixed Effects | | Controlled | | | | | | |

*, **, and *** indicate statistical significance at the 10%, 5%, and 1% levels, respectively. Standard errors are reported in parentheses below the coefficients.

measure (degree of financial vulnerability), results across all regions remain consistent with the baseline findings, with the strongest mitigating effect of Internet use in western regions, followed by eastern regions, and the weakest effect in central regions.

Regarding urban-rural heterogeneity, for the binary indicator of household financial vulnerability, Internet use significantly reduces the probability of vulnerability only in rural areas, while no significant impact is observed in urban areas. For the degree of vulnerability, results align consistently with baseline outcomes, demonstrating that Internet use exhibits a stronger mitigating effect in rural households compared to urban households.

Considering heterogeneity by household-head age, in the case of the binary indicator, Internet use significantly decreases the likelihood of household financial vulnerability only for households headed by individuals under the age of 60; no significant effect is found for households with heads aged 60 and above. For the continuous measure, Internet use consistently lowers the degree of household financial vulnerability across age groups, but the magnitude of this effect is notably greater in households headed by younger individuals (under the age of 60) than in older-headed households (aged 60 and above).

## 5. Discussion

The empirical findings of this study robustly demonstrate that Internet use serves as a significant mitigating factor against household financial vulnerability in China, both in terms of its likelihood and severity. This section delves into the broader implications of these findings, interpreting them in the context of existing theoretical frameworks and prior empirical work, and deriving actionable policy insights.

### 5.1. Comparison with past findings

Our results corroborate and extend a growing body of literature on the positive socioeconomic impacts of digital technologies. The finding that Internet use enhances household income aligns with previous studies [15,30], confirming the role of digital access in improving labor market outcomes and entrepreneurial opportunities. Similarly, the positive association between Internet use and wealth accumulation, particularly in financial assets, highlights the Internet's function in reducing information asymmetry and transaction costs in financial markets [33–34].

                                                                   

However, our study moves beyond these established pathways by introducing and empirically validating a more comprehensive framework. While prior research often focused on isolated aspects like income or insurance participation, we simultaneously model three core mechanisms (income growth, wealth accumulation, and risk management optimization), providing a more holistic understanding of how digital penetration translates into enhanced financial resilience. Furthermore, our approach to measuring financial vulnerability represents a significant advancement. Unlike studies that employed binary indicators [17–18] or simple ordinal categories, our continuous measure, incorporating both financial margin and solvency, captures not just the incidence but also the intensity of financial distress. This allows for a more nuanced estimation of the Internet's impact, revealing that it not only reduces the probability of falling into vulnerability but also significantly lessens its depth for those who are vulnerable.

The heterogeneity analysis yields crucial insights that both complement and complicate the narrative of the digital divide. The stronger effects observed in western regions and rural areas suggest that Internet access provides a "latecomer advantage" or a "leapfrogging" opportunity for historically disadvantaged households. This finding is consistent with the concept of diminishing marginal returns; where traditional financial infrastructure is weakest, the marginal utility of digital connectivity is highest [14]. Conversely, the weaker effect among households with older heads underscores that the "second-level digital divide" (differences in use) remains a formidable challenge, even as access gaps narrow. This indicates that the benefits of the Internet are not automatic but are mediated by an individual's capacity to effectively utilize the technology.

### 5.2. Theoretical and policy implications

Theoretically, our findings underscore the conceptualization of Internet proficiency as a form of "digital capital" [42]. It is not merely a tool but a transformative asset that empowers households to build human capital, expand social networks, access markets, and optimize financial decisions. This study integrates elements from information economics, household finance, and risk management theory, demonstrating how digital capital interacts with traditional forms of capital to determine a household's financial robustness. These insights translate into several concrete policy implications.

Deepening Digital Infrastructure with a Pro-Equity Focus. Policy efforts must extend beyond blanket broadband rollout. Targeted investments are needed to ensure high-quality, affordable Internet access in western and rural regions. This addresses the "access divide" and maximizes the societal returns on digital investment by reaching populations where the marginal benefit is greatest.

Launching Digital Literacy Programs for Vulnerable Demographics. Bridging the "usage divide" is imperative. Public initiatives should aim to enhance digital skills, particularly among older adults and less-educated populations. Training should go beyond basic operation to include practical applications in personal finance, online banking, accessing e-government services, and identifying misinformation, thereby enabling them to convert digital access into tangible financial benefits.

Fostering Innovation in Fintech and Digital Financial Products. Regulators and financial institutions should encourage the development of inclusive, user-friendly digital financial products. This includes micro-insurance products tailored for low-income households, simplified digital investment platforms, and fintech solutions that can leverage alternative data for credit scoring. This helps channel the benefits of Internet use directly into improved risk management for families.

Implementing Differentiated and Precise Policy Interventions. Policymakers should reject a one-size-fits-all approach. In less developed regions, policy should focus on building foundational digital infrastructure and literacy. In more advanced areas, the focus can shift to promoting advanced digital financial services and protecting consumers from digital risks. For the elderly, policies must be designed to ensure inclusion, providing tailored support to help them navigate the digital economy safely and effectively.

## 6. Conclusions and prospects

### 6.1. Key conclusions

This study utilizes nationally representative panel data from the CFPS (2016, 2018, 2020) to empirically investigate the impact of Internet use on household financial vulnerability and its underlying mechanisms. The main conclusions are as follows.

First, Internet use significantly mitigates both the probability and the degree of household financial vulnerability. This core finding remains robust after a series of stringent tests, including alternative measures of key variables, different model specifications, and corrections for endogeneity using an instrumental variable approach. Second, the mechanism analysis reveals that Internet use alleviates financial vulnerability primarily through three channels: promoting household income growth, facilitating wealth accumulation, and optimizing risk management capabilities. This multi-pathway framework provides a comprehensive explanation for the observed effect. Third, significant heterogeneity exists in the impact of Internet use. The mitigating effects are particularly pronounced among households in western China, rural areas, and those headed by individuals under the age of 60. These findings highlight the unequal distribution of digital dividends and underscore the existence of the "usage divide" alongside the "access divide".

### 6.2. Limitations

Despite its contributions, this study is subject to several limitations that warrant acknowledgment. First, although we employed panel data, the study period concludes in 2020. The rapid evolution of digital technologies means that our findings may not fully capture the most recent dynamics of Internet use and its financial implications. Future research with more contemporary data could yield additional insights. Second, while we endeavored to construct a precise measure of financial vulnerability, certain aspects, such as the definition of "unexpected expenditures" may not encompass all potential financial shocks a household might face. The measurement of some mediating variables could also be further refined. Third, although we implemented an IV strategy to address endogeneity, the quest for a perfectly exogenous instrument is perpetual. While geographically determined distance to a digital hub is plausibly exogenous, we cannot entirely rule out the possibility of unobserved confounding factors that might correlate with both the instrument and the error term.

### 6.3. Future research directions

Based on these findings and limitations, we propose several promising avenues for future research. First, future studies could disaggregate "Internet use" into different types to examine whether specific online activities have divergent effects on financial outcomes. This would provide more granular guidance for policy. Second, while this study identifies associated mechanisms, the precise behavioral and psychological channels remain a "black box". Integrating experimental methods or detailed survey modules on behavioral traits could unpack these mechanisms more effectively. Third, research could explore the impact of the next generation of digital technologies on household financial vulnerability, building upon the foundation established here. Finally, future work could conduct cross-country comparative analyses to examine how the relationship between Internet use and financial vulnerability varies across different institutional and regulatory environments, testing the generalizability of our findings.

### Acknowledgments

In the Dependent variable: Household financial vulnerability section, total income includes wage income, business income, financial management income, asset income, and household transfers; easily current assets include cash and deposits; regular expenditures include expenses on food, clothing, housing, daily necessities, transportation, communication, entertainment, mortgages, and vehicle loans; unexpected expenditures include medical expenses and transfer payments. In the Independent variable: Internet use section, CFPS 2020 questionnaire differs slightly, and adjustments were made for consistency. The highest frequency of Internet use among household members is used to represent the household's Internet use level. In the Control variables section, CFPS includes items assessing respondents' perceptions of the severity of issues in environmental protection, income inequality, employment, education, healthcare, housing, social security, and government integrity on a scale from 0 (not severe) to 10 (very severe). Scores across these eight areas are summed to represent overall social environment conditions. In the Replacing dependent variables section, household

assets include land, real estate, financial and fixed assets; household liabilities include housing loans and other debts; household income comprises wage income, business income, property income, transfer income, and other miscellaneous income; household expenditures include consumption expenses, transfer payments, welfare expenditures, and housing loan repayments. In the Income and wealth growth mechanism section, all income and wealth variables are transformed using logarithms.

In the Optimized risk management mechanism section, annual household expenditure on social ceremonies is used as a proxy for household social capital, measured in log form. Due to limited observations of financial market participation, cross-sectional data from three surveys are used for regression.

## Author contributions

**Conceptualization:** Zeliang Yu.

**Data curation:** Lin He.

**Formal analysis:** Pei Guo.

**Funding acquisition:** Zeliang Yu.

**Methodology:** Heyu Li.

**Software:** Lin He.

**Supervision:** Pei Guo.

**Validation:** Heyu Li.

**Writing – original draft:** Zeliang Yu.

**Writing – review & editing:** Heyu Li.

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
