## [Decision Letter · Decision Letter 0]

18 Aug 2025

PONE-D-25-20548Does Internet Use Alleviate Household Financial Vulnerability? An Empirical Analysis Based on Panel Data from China Family Panel Studies (CFPS)PLOS ONE?

Dear Dr. Li,

Thank you for submitting your manuscript to PLOS ONE. After careful consideration, we feel that it has merit but does not fully meet PLOS ONE’s publication criteria as it currently stands. Therefore, we invite you to submit a revised version of the manuscript that addresses the points raised during the review process.

**For detailed comments, please refer to the recommendations provided at the end of this email.**

We look forward to receiving your revised manuscript.

Kind regards,

Imran Ur Rahman, Ph.D

Academic Editor

PLOS ONE

**Journal Requirements:**

1. When submitting your revision, we need you to address these additional requirements. Please ensure that your manuscript meets PLOS ONE's style requirements, including those for file naming. The PLOS ONE style templates can be found at https://journals.plos.org/plosone/s/file?id=wjVg/PLOSOne_formatting_sample_main_body.pdf and https://journals.plos.org/plosone/s/file?id=ba62/PLOSOne_formatting_sample_title_authors_affiliations.pdf 2. Thank you for stating the following financial disclosure: a. National Key R&D Program “Intergovernmental International Science and Technology Innovation Cooperation” project "Natural Solutions-based Agricultural Nutrient Management and Sustainable Transformation between China and the European Union" (2023YFE0105000);b. National Natural Science Foundation Youth Project “Research on Resilience Governance of Poverty-stricken Farmers in Western China under the Impact of Natural Disasters” (72403234);c. Guangdong Provincial Federation of Social Sciences “Zhongkai Agricultural and Engineering University Guangdong-Hong Kong-Macao Greater Bay Area Rural Finance and Agricultural Investment Research Center” project (Guangdong Federation of Social Sciences Letter [2022] No. 5).   Please state what role the funders took in the study.  If the funders had no role, please state: "The funders had no role in study design, data collection and analysis, decision to publish, or preparation of the manuscript." If this statement is not correct you must amend it as needed. Please include this amended Role of Funder statement in your cover letter; we will change the online submission form on your behalf. 3. Thank you for stating the following in the Acknowledgments Section of your manuscript: a. In 3.2.1 Dependent Variable: Household Financial Vulnerability section, total income includes wage income, business income, financial management income, asset income, and household transfers; easily current assets include cash and deposits; regular expenditures include expenses on food, clothing, housing, daily necessities, transportation, communication, entertainment, mortgages, and vehicle loans; unexpected expenditures include medical expenses and transfer payments.b. In 3.2.2 Independent Variable: Internet Use section, CFPS 2020 questionnaire differs slightly, and adjustments were made for consistency. The highest frequency of Internet use among household members is used to represent the household’s Internet use level.c. In 3.2.3 Control Variables section, CFPS includes items assessing respondents’ perceptions of the severity of issues in environmental protection, income inequality, employment, education, healthcare, housing, social security, and government integrity on a scale from 0 (not severe) to 10 (very severe). Scores across these eight areas are summed to represent overall social environment conditions.d. In 4.2.1 Replacing Dependent Variables section, household assets include land, real estate, financial and fixed assets; household liabilities include housing loans and other debts; household income comprises wage income, business income, property income, transfer income, and other miscellaneous income; household expenditures include consumption expenses, transfer payments, welfare expenditures, and housing loan repayments.e. In 4.4.1 Income and Wealth Growth Mechanism section, all income and wealth variables are transformed using logarithms.f. In 4.4.2 Optimized Risk Management Mechanism section, annual household expenditure on social ceremonies is used as a proxy for household social capital, measured in log form. Due to limited observations of financial market participation, cross-sectional data from three surveys are used for regression. We note that you have provided funding information that is not currently declared in your Funding Statement. However, funding information should not appear in the Acknowledgments section or other areas of your manuscript. We will only publish funding information present in the Funding Statement section of the online submission form. Please remove any funding-related text from the manuscript and let us know how you would like to update your Funding Statement. Currently, your Funding Statement reads as follows: a. National Key R&D Program “Intergovernmental International Science and Technology Innovation Cooperation” project "Natural Solutions-based Agricultural Nutrient Management and Sustainable Transformation between China and the European Union" (2023YFE0105000);b. National Natural Science Foundation Youth Project “Research on Resilience Governance of Poverty-stricken Farmers in Western China under the Impact of Natural Disasters” (72403234);c. Guangdong Provincial Federation of Social Sciences “Zhongkai Agricultural and Engineering University Guangdong-Hong Kong-Macao Greater Bay Area Rural Finance and Agricultural Investment Research Center” project (Guangdong Federation of Social Sciences Letter [2022] No. 5).  Please include your amended statements within your cover letter; we will change the online submission form on your behalf. 4. In the online submission form, you indicated that your data is available only on request from a third party. Please note that your Data Availability Statement is currently missing the contact details for the third party, such as an email address or a link to where data requests can be made. Please update your statement with the missing information. 5. If the reviewer comments include a recommendation to cite specific previously published works, please review and evaluate these publications to determine whether they are relevant and should be cited. There is no requirement to cite these works unless the editor has indicated otherwise. 

**Additional Editor Comments:**

The topic of the research has some possible implications, but there are major points that need to be addressed. I hope the authors will consider the following suggestions and enhance the overall outlook and framework of the research:

1. The introduction section should be further enhanced by elaborating on the key gaps and contributions. I would suggest the authors enhance the theoretical argument in the introduction section.

2. The literature review needs to be enhanced, and up-to-date literature studies should be added to motivate the readers with the background and currency of the subject of the research.

3. The methodology section is good and used different approaches; however, if possible, additional diagnostic tests, including correlations, etc., may further validate the results.

4. Please further explain and justify the method used.

5. The authors should enhance their discussion section as a separate section. Please provide theoretical and practical policy implications and compare the outcomes to past findings.

6. The conclusion section needs to be modified. I would recommend adding the key outcomes, followed by limitations and future research directions.

7. The organization and formatting of the article require modifications in accordance with standard research design and framework. The authors should keep a uniform formatting for the text and tables/figures throughout the manuscript and follow the guidelines of the journal.

8. There are spelling, grammar, and phrasing issues in the manuscript. The authors should revise the language through proofreading.

Reviewers' comments:

**Comments to the Author**

1. Is the manuscript technically sound, and do the data support the conclusions?

Reviewer #1: Yes

Reviewer #2: Partly

2. Has the statistical analysis been performed appropriately and rigorously?

Reviewer #1: Yes

Reviewer #2: Yes

3. Have the authors made all data underlying the findings in their manuscript fully available?

Reviewer #1: Yes

Reviewer #2: Yes

4. Is the manuscript presented in an intelligible fashion and written in standard English?

Reviewer #1: Yes

Reviewer #2: Yes

**Reviewer #1:**  The study is well-structured, data-rich, and covers an important topic. However, it must make significant improvements to its empirical design, theoretical framing, and external validity before meeting publication standards. Thus, a MAJOR REVISION must be made. Once these issues are resolved, its contribution will be significantly stronger and more persuasive.

Dear authors,

Kindly, see my comments and revise your paper.

Good Luck!

**Reviewer #2:** 1. Lack of Citations:

A major concern is the absence of citations throughout many paragraphs, which weakens the manuscript’s academic integrity and grounding. The author must ensure that every key claim, concept, or data point is supported by appropriate and recent references. Proper citation not only situates the study within the existing literature but also enhances the reliability and scholarly value of the work. A thorough literature review with accurate citations is fundamental for strengthening the argument and demonstrating awareness of relevant research.

2. Theoretical Analysis:

It is imperative to explicitly define and discuss the key theories underpinning the study, thereby providing a solid conceptual foundation. Clear articulation of theoretical perspectives will enhance the coherence of the research model and help readers understand the rationale behind variable selection and hypothesized relationships.

3. Transition from Theory to Hypotheses:

The shift from theoretical discussion to hypothesis formulation is abrupt and lacks a smooth, logical flow. To improve readability and conceptual clarity, the author should establish clear linkages between theoretical constructs and hypotheses, ensuring each hypothesis is clearly grounded in the preceding theory. This will make the manuscript more structured and persuasive, guiding the reader seamlessly through the research design.

4. Justification for Variable Selection:

There is an absence of clear justification for why the main variables were selected for this study. The author should provide a detailed rationale for the inclusion of each variable, grounded in theory and supported by prior empirical evidence. Clarifying the relevance and significance of these variables will strengthen the manuscript’s conceptual framework and enhance the study’s contribution to the field.

5. Validity Testing of Control Variables:

Control variables are incorporated in the analysis but their validity is neither tested nor discussed. It is critical to assess the appropriateness and impact of control variables based on established guidelines such as those proposed by Memo et al. (2024) and Becker et al. (2015). Testing and reporting the validity of control variables will improve the robustness of the results and ensure that the effects of confounding factors are properly accounted for.

6. Cross-Sectional Dependence Tests:

The manuscript lacks any mention of cross-sectional dependence tests, which is a significant methodological omission, especially in studies involving panel data. The author should conduct appropriate tests to detect cross-sectional dependence and clearly explain how any identified issues are addressed. This step is vital to ensure the reliability and validity of the empirical results and to avoid biased inferences.

7. Limitations and Future Research:

A notable gap in the manuscript is the absence of a discussion on the study’s limitations and directions for future research. Including a dedicated section that critically acknowledges the study’s constraints and proposes areas for further investigation will not only add depth to the research but also guide subsequent studies. This reflection demonstrates scholarly rigor and openness to ongoing inquiry, enhancing the overall contribution of the manuscript.

**Do you want your identity to be public for this peer review?** For information about this choice, including consent withdrawal, please see our Privacy Policy

Reviewer #1: **Yes:** Esraa Alharasis

Reviewer #2: No

---

## [Author Response · Author response to Decision Letter 1]

19 Sep 2025

Response to Editors and Reviewers

Dear Editor and Reviewers,

We sincerely thank the Academic Editor and reviewers for their constructive comments and suggestions, which have greatly helped us improve the quality and clarity of our manuscript. We have carefully revised the manuscript in accordance with the feedback and the journal’s requirements. Below, we provide a detailed, point-by-point response.

Editor’s Comments

Comment 1: The introduction section should be further enhanced by elaborating on the key gaps and contributions. I would suggest the authors enhance the theoretical argument in the introduction section.

Response: We would like to kindly note that in the original version of the introduction, we had already included a preliminary statement regarding the contributions of this study as follows:

Firstly, utilizing three-wave CFPS panel data to better estimate and identify causal relationships between Internet use and household financial vulnerability, providing novel insights into understanding China’s household financial vulnerability.

Secondly, expanding the analytical framework by examining mechanisms through income growth, wealth accumulation, and risk management optimization, thereby contributing to a comprehensive understanding of digital technology’s value.

Thirdly, conducting heterogeneity analyses across regional, urban-rural, and household characteristics to provide empirical evidence supporting precise policy formulation.

In direct response to your suggestions, we have substantially revised the introduction to provide a more explicit and systematic elaboration of the research gaps, theoretical underpinnings, and contributions of our study. Specifically, we have:

Clearly identified key gaps in the existing literature, particularly the lack of a multi-mechanism framework, insufficient handling of endogeneity, limited heterogeneity analysis, and oversimplified measurement of financial vulnerability.

Strengthened the theoretical argument by introducing a coherent mediating mechanism framework through which Internet use affects household financial vulnerability—namely, via income growth, wealth accumulation, and risk management optimization.

Explicitly outlined the marginal contributions of our study in methodological, theoretical, and policy-oriented terms, ensuring they are directly aligned with the identified research gaps.

We believe these revisions have significantly enhanced the clarity, theoretical depth, and overall rigor of the introduction. The revised version now provides a stronger foundation for the empirical analysis that follows.

Comment 2: The literature review needs to be enhanced, and up-to-date literature studies should be added to motivate the readers with the background and currency of the subject of the research.

Response: We are grateful for this suggestion. In direct response, we have comprehensively enhanced the literature review component within the introduction to better establish the study's background and currency. Specifically, we have:

Integrated a broader and more up-to-date range of literature: We have incorporated seminal works on the conceptualization of household financial vulnerability (e.g. Loschiavo et al.,2025; Anderloni et al., 2012) and significantly expanded the discussion on digital technology's impact with recent, high-quality empirical studies from China (e.g. Kouladoum, 2025 on human capital; Yuan et al, 2024 on income; Wang et al., 2023 on household financial assets; Zhu et al., 2022 on the digital divide; Yin et al., 2022 on insurance participation). This provides a more robust and contemporary scholarly foundation.

Strengthened the narrative to motivate the research: The enhanced literature review is now woven into a logical narrative that not only summarizes existing findings but also critically synthesizes them to clearly highlight the evolving research landscape and the specific, timely gaps that our study addresses. This effectively motivates the reader by underscoring the necessity and relevance of our research questions.

Ensured seamless integration within the PLOS ONE format: Following the journal's style, we have consciously avoided a separate literature review section. Instead, we have meticulously synthesized the reviewed literature throughout the introduction, ensuring it serves to naturally introduce the background, establish currency, and lead into the study's objectives without disrupting the manuscript's flow and conciseness.

We believe these revisions have successfully enriched the background context and scholarly motivation of the study, providing readers with a clearer and more compelling understanding of the research's timeliness and value.

Comment 3: The methodology section is good and used different approaches; however, if possible, additional diagnostic tests, including correlations, etc., may further validate the results.

Response: We thank the reviewer for this positive feedback and constructive suggestion. We have conducted additional diagnostic tests to further validate our results, as detailed below:

Multicollinearity Diagnosis: We calculated Variance Inflation Factors (VIF) for all independent variables. Results indicate no concerning multicollinearity, with a mean VIF of 1.44 and all individual VIF values well below the common threshold of 10 (the highest being 2.13 for the variable Region). This ensures the stability and precision of our coefficient estimates. Detailed results are presented in the last paragraph of the Benchmark regression section.

Enhanced Instrumental Variable (IV) Diagnostics: We have strengthened our IV analysis by: (1) explicitly reporting the first-stage coefficient for our instrument (Distance → Internet Use), which shows a statistically significant negative relationship (p < 0.05), empirically validating the relevance condition; (2) reporting the second-stage coefficient for the key variable (Internet Use → Financial Vulnerability), which remains significantly negative after controlling for endogeneity; and (3) discussing the results of the Wald test of exogeneity (χ²(1) = 16.64, p = 0.0000), which formally rejects the null hypothesis that Internet use is exogenous, thus justifying our IV strategy.

Heteroscedasticity Robustness Check: To address potential heteroscedasticity in our Tobit model estimates of financial vulnerability degree, we re-estimated the model using bootstrapped standard errors (500 replications). Results confirm that all coefficients maintained their significance, direction, and magnitude compared to original estimates. For example, Internet use retained strong statistical significance (p < 0.01) with stable coefficient estimates and similar standard error magnitudes. This confirms that heteroscedasticity does not materially affect our inferences, and we therefore retain the original specification in the main manuscript for consistency and clarity.

Proactive Methodological Enhancement: Beyond addressing reviewer comments, we have added a new subsection (Robustness Checks) to the Statistical Methods section. This addition outlines our comprehensive robustness strategies (including measure replacement and Heckman correction models) and provides readers with a clear roadmap to our robustness validation, thereby enhancing the manuscript's methodological transparency and logical flow.

We believe these additions significantly strengthen the empirical rigor of our study.

Comment 4: Please further explain and justify the method used.

Response: We appreciate this suggestion. We have thoroughly revised the methodology section to provide a more comprehensive explanation and justification for our choice of econometric models and empirical strategies:

Enhanced Justification for Model Selection: (1) For the Fixed-Effects Panel Logit model: We explicitly state that it was chosen over a Linear Probability Model (LPM) to avoid predicting probabilities outside the [0,1] range and to address heteroscedastic errors. Most importantly, we emphasize its capability to control for unobserved, time-invariant household heterogeneity (e.g. innate risk preferences, cultural attitudes), which is a likely source of omitted variable bias. (2) For the Panel Tobit model: We clarify that it is specifically designed to handle our left-censored dependent variable (financial vulnerability degree censored at 1). We explain that applying OLS to such data would yield inconsistent estimates, and the Tobit model provides a theoretically correct framework for obtaining consistent parameters.

Clarification on Variable Selection Approach: Our control variables were chosen a priori based on economic theory and established literature on household financial vulnerability. We have now added a justification in the control variables subsection, explaining that the selected covariates (e.g. household head's demographics, household characteristics, regional factors) are fundamental determinants empirically proven to influence financial health, ensuring our model is both theoretically grounded and economically interpretable.

Deeper Discussion of Endogeneity and IV Choice: (1) We expanded the discussion on potential endogeneity (reverse causality, omitted time-varying variables) to robustly justify the need for an Instrumental Variable (IV) approach. (2) We provided a more nuanced theoretical justification for our instrument (distance to Hangzhou). We argue that, after controlling for provincial GDP and regional fixed effects, this geographical instrument is plausibly exogenous and likely affects financial vulnerability only through its impact on Internet access quality (relevance condition), thus satisfying the exclusion restriction.

We believe these revisions offer a clearer, more compelling, and accurate rationale for our empirical strategy, demonstrating that our methods were carefully chosen to address the specific statistical challenges and causal identification problems inherent in our research questions.

Comment 5: The authors should enhance their discussion section as a separate section. Please provide theoretical and practical policy implications and compare the outcomes to past findings.

Response: We are deeply grateful to the editor for this insightful suggestion. In direct response, we have now added a standalone section titled “Discussion” to the manuscript. This new section comprehensively addresses the editor's request by:

Comparing our outcomes with past findings: We explicitly discuss how our results confirm prior research on Internet use and income/wealth, while also highlighting our novel contributions—specifically, our multi-mechanism framework and more precise measurement of financial vulnerability that captures both incidence and intensity.

Elaborating on theoretical implications: We interpret our findings through the lens of "digital capital," explaining how Internet use acts as a transformative asset that empowers households and integrates theories from information economics and household finance.

Providing detailed practical policy implications: We moved beyond general statements to propose four specific, actionable policy recommendations focused on pro-equity infrastructure, digital literacy for vulnerable groups, fintech innovation, and differentiated policy interventions.

We believe this new section significantly strengthens the manuscript by deeply contextualizing our findings within the broader scholarly discourse and extracting clear, practical value from our research.

Comment 6: The conclusion section needs to be modified. I would recommend adding the key outcomes, followed by limitations and future research directions.

Response: We sincerely thank the editor for this crucial recommendation. We have comprehensively revised the original section into a new section Conclusion and Prospects, which now strictly follows the suggested structure:

Key Outcomes: We begin the section with a succinct and bullet-point-style summary of the three most critical findings from our research, providing readers with a clear take-home message.

Limitations: We openly and honestly discuss the limitations of our study, focusing on data timeliness, measurement challenges, and persistent concerns regarding endogeneity, thereby demonstrating a nuanced understanding of our study's boundaries.

Future Research Directions: We propose several concrete and promising pathways for future scholarship, including investigating types of Internet use, exploring behavioral mechanisms, and studying emerging technologies. This moves the field forward by identifying clear next steps.

We believe these revisions enhance the academic rigor and completeness of our manuscript, providing a balanced and forward-looking conclusion.

Comment 7: The organization and formatting of the article require modifications in accordance with standard research design and framework. The authors should keep a uniform formatting for the text and tables/figures throughout the manuscript and follow the guidelines of the journal.

Response: We sincerely appreciate this valuable suggestion. In the revised version, we have carefully reorganized sections to align more closely with standard research design and framework, ensuring a clearer logical flow between the introduction, methodology, results, and discussion. In addition, we have applied uniform formatting across all parts of the manuscript, including consistency in headings, references and tables. All elements have been revised to strictly follow the PLOS ONE formatting guidelines, as recommended. We believe these changes have enhanced the overall readability and professionalism of the paper.

Comment 8: There are spelling, grammar, and phrasing issues in the manuscript. The authors should revise the language through proofreading.

Response: Thank you very much for highlighting this important issue. We have conducted a thorough language revision of the entire manuscript. The spelling errors, grammatical inconsistencies, and awkward expressions have been corrected. To ensure clarity and precision, the manuscript was carefully proofread by multiple team members and cross-checked using professional editing tools. We have also refined phrasing in key sections such as the abstract, methods and results to improve readability and maintain an academic tone throughout. We believe these revisions substantially improve the clarity and overall presentation of the manuscript.

Reviewer #1

We would like to note that, as clarified by the Senior Editor, the comments from Reviewer #1 are not related to our submission. In line with the Editor’s instructions, we have not revised the manuscript in response to these comments. We appreciate the clarification and thank the Editor for resolving this matter.

Reviewer #2

Comment 1. Lack of Citations: A major concern is the absence of citations throughout many paragraphs, which weakens the manuscript’s academic integrity and grounding. The author must ensure that every key claim, concept, or data point is supported by appropriate and recent references. Proper citation not only situates the study within the existing literature but also enhances the reliability and scholarly value of the work. A thorough literature review with accurate citations is fundamental for strengthening the argument and demonstrating awareness of relevant research.

Response: We sincerely thank the reviewer for this critical and constructive feedback. We fully agree that comprehensive citations are fundamental to academic integrity and for situating one's work within the existing literature. We have thoroughly reviewed the manuscript and added numerous citations to strengthen the scholarly foundation of our arguments, particularly in the Introduction and Theoretical Analysis sections. Key additions and revisions include:

In the Introduction section, we have integrated a broader and more up-to-date range of literature to substantiate our claims. This includes incorporating seminal works on the conceptualization of household financial vulnerability (e.g., Anderloni et al., 2012; Loschiavo, 2025) and significantly expanding the discussion on digital technology's impact with recent, high-quality empirical studies from China. These new citations cover critical areas such

---

## [Decision Letter · Decision Letter 1]

11 Jan 2026

Does Internet use alleviate household financial vulnerability? An empirical analysis based on panel data from China Family Panel Studies (CFPS)PLOS One?

Dear Dr. Li,

Thank you for submitting your manuscript to PLOS ONE. After careful consideration, we feel that it has merit but does not fully meet PLOS ONE’s publication criteria as it currently stands. Therefore, we invite you to submit a revised version of the manuscript that addresses the points raised during the review process.

We look forward to receiving your revised manuscript.

Kind regards,

Imran Ur Rahman, Ph.D

Academic Editor

PLOS One

Journal Requirements:

Additional Editor Comments:

The authors have addressed the majority of the comments. Thank you for modifying the manuscript in accordance with the recommendations of the esteemed reviewers. However, before moving ahead with the acceptance, there are minor issues that require attention:

1) The authors should add an introductory sentence or two at the start of the abstract.

2) Please add the keywords after the abstract. The authors have removed the keywords in the revised version.

3) If possible, please add a figure highlighting the research design/framework.

4) The authors have removed the numberings. Please add numberings to the headings and subheadings.

5) There are minor grammatical and phrasing issues. The authors should revise the language through proofreading.

Reviewers' comments:

Reviewer's Responses to Questions

**Comments to the Author**

Reviewer #1: All comments have been addressed

Reviewer #2: All comments have been addressed

2. Is the manuscript technically sound, and do the data support the conclusions?

Reviewer #1: Yes

Reviewer #2: Yes

3. Has the statistical analysis been performed appropriately and rigorously?

Reviewer #1: Yes

Reviewer #2: Yes

4. Have the authors made all data underlying the findings in their manuscript fully available?

Reviewer #1: Yes

Reviewer #2: Yes

5. Is the manuscript presented in an intelligible fashion and written in standard English?

Reviewer #1: Yes

Reviewer #2: Yes

Reviewer #1: Dear authors,

I have carefully reviewed the revised version of the manuscript and am satisfied with the corrections and clarifications made by the authors. The revised paper has addressed the previous comments adequately, and I find it suitable for publication in its current form. Thus, I accept the paper.

Good Luck!

Reviewer #2: (No Response)

**Do you want your identity to be public for this peer review?** For information about this choice, including consent withdrawal, please see our Privacy Policy

Reviewer #1: No

Reviewer #2: No

---

## [Author Response · Author response to Decision Letter 2]

13 Jan 2026

Dear Editor and Reviewers,

Thank you for your careful assessment of our revised manuscript entitled “Does Internet use alleviate household financial vulnerability? An empirical analysis based on panel data from China Family Panel Studies (CFPS)” (Manuscript ID: PONE-D-25-20548). We appreciate your constructive guidance and are pleased to submit a further revised version that addresses the remaining minor issues prior to acceptance.

In this revision, we have implemented the following updates: (1) added an introductory sentence at the start of the abstract; (2) reinstated and provided the keywords immediately after the abstract; (3) added a figure illustrating the mechanism analysis; (4) restored numbering for all headings and subheadings; and (5) proofread the manuscript to correct minor grammatical and phrasing issues.

We respond to each point below:

Editor Comment 1) The authors should add an introductory sentence or two at the start of the abstract.

Response: We have revised the abstract by adding an introductory sentence at the beginning to better motivate the research question and contextualize the study.

Change made: An opening sentence has been inserted at the start of the Abstract section.

Editor Comment 2) Please add the keywords after the abstract. The authors have removed the keywords in the revised version.

Response: We have added the keywords immediately after the abstract, as requested.

Change made: We inserted the keywords in the manuscript right after the abstract. The keywords are:

Keywords: Internet use; Household financial vulnerability; Income growth; Wealth accumulation; Risk management capability.

Editor Comment 3) If possible, please add a figure highlighting the research design/framework.

Response: We agree that a visual summary improves clarity. Accordingly, we have added a new figure that highlights the overall research design and analytical framework.

Change made: We added Figure 1, titled “Flowchart of Mechanism Analysis”, along with an explanatory caption in the manuscript.

Editor Comment 4) The authors have removed the numberings. Please add numberings to the headings and subheadings.

Response: We have reinstated numberings for all headings and subheadings to improve readability and navigation.

Change made: Section and subsection numberings (e.g., 1, 1.1, 1.2, …) has been restored throughout the manuscript.

Editor Comment 5) There are minor grammatical and phrasing issues. The authors should revise the language through proofreading.

Response: We have carefully proofread the entire manuscript and corrected minor grammatical errors, awkward phrasing, and inconsistencies in style and terminology.

Change made: Language edits have been implemented across the manuscript to improve clarity and academic tone while preserving the original meaning.

Reviewer Comments

We also note that Reviewer #1 indicated satisfaction with the revisions and recommended acceptance without raising any further specific comments requiring a point-by-point response. Reviewer #2 provided no additional response in this round; therefore, there were no reviewer-specific comments for us to address beyond the editorial requests above.

Thank you again for your time and consideration. We hope that these revisions fully address the remaining issues, and we would be grateful for your confirmation that the manuscript is now suitable for acceptance.

Sincerely,

Heyu Li (on behalf of all authors)

heyu.li@foxmail.com

---

## [Editor Report · Decision Letter 2]

8 Feb 2026

Does Internet use alleviate household financial vulnerability? An empirical analysis based on panel data from China Family Panel Studies (CFPS)

PONE-D-25-20548R2

Dear Dr. Li,

We’re pleased to inform you that your manuscript has been judged scientifically suitable for publication and will be formally accepted for publication once it meets all outstanding technical requirements.

Kind regards,

Imran Ur Rahman, Ph.D

Academic Editor

PLOS One

Additional Editor Comments (optional):

The authors have addressed all the issues. It is recommended the authors further enhance the formatting of the tables and equations.
---

## [Editor Report · Acceptance letter]

PONE-D-25-20548R2

PLOS One

Dear Dr. Li,

I'm pleased to inform you that your manuscript has been deemed suitable for publication in PLOS One. Congratulations! Your manuscript is now being handed over to our production team.

Kind regards,

on behalf of

Dr. Imran Ur Rahman

Academic Editor

PLOS One